# Numerical Testing of Switch Point Dynamics—A Curved Beam with a Variable Cross-Section

**DOI:** 10.3390/ma13030701

**Published:** 2020-02-04

**Authors:** Jerzy Kisilowski, Rafał Kowalik

**Affiliations:** 1Faculty of Transport, Electrical Engineering and Computer Science, University of Technology and Humanities, ul. Malczewskiego 29, 26-600 Radom, Poland; 2Department of Avionics and Control Systems, Polish Air Force University, ul. Dywizjonu 303 nr 35, 08-521 Deblin, Poland

**Keywords:** vibration, railways, switch point, beam with/of a variable cross-section, turnout

## Abstract

The article presents mathematical considerations on the dynamics of the springing switch point being an element of the railway junction. Due to the structure of the switch point, mathematical analysis was divided into two stages: The first stage refers to the analysis of the dynamics of the switch point as a beam of variable rectilinear stiffness to which three forces (coming from three closures of switch drives) placed in the initial section of the switch point are applied. The next stage of the analysis concerns an identical beam, but curved, with a variable cross-section. In both cases, the beam is subjected to a vertical force resulting from forces from the rail vehicle. The calculations refer to a switch point of 23 m length and a curvature radius *R* = 1200 m. The first stage of the switch point analysis refers to the movement of a rail vehicle on a straight track, and the second stage concerns the rail vehicle movement on a reverse path. This article also provides an analysis of mode vibrations of a curved beam with a variable cross-section, and variable inertia and stiffness moments (further in the article the changes will be referred to as beam parameter changes). It is assumed that the beam is loaded with vertical forces (coming) from a rail vehicle. The solution was found by applying the Ritz method, which served to present the fourth-order partial equations as ordinary differential ones. The numerical research whose results are given aimed to define how the changes in beam parameters and vertical load affect mode vibrations of the beam.

## 1. Introduction

Free vibrations of continuous mechanical systems, which include a switch point—a rail turnout element—do not solely depend on a load model (conservative or non-conservative one) or a type of support structures, but also on the change in both the cross-section, inertia moment and stiffness along its length. An additional quantity to consider in mathematical analysis is a switch point curve.

The dynamics of a nonlinear beam on a linear elastic foundation was tested by Akour [1], who investigated a simply supported beam with harmonic distributed load, on a one-parameter elastic foundation with damping included. The impact of damping coefficient, beam nonlinearity, and vibration frequency on beam stability was tested. An issue of nonlinear vibrations for Euler–Bernoulli beams was presented in [2]. Two methods were used to determine the frequency of vibrations: the variation iteration method (VIM) and the parameterized perturbation method (PPM). Both methods allow obtaining results without an introduction of a small amplitude parameter to the main equation that describes transverse vibrations of the beam.

The perturbation method was also used in paper [3]. They focused on a beam with a concentrated load perpendicular to its longitudinal axis, when the beam is located on a rigid one-parameter elastic foundation. Apart from the analysis of transverse vibrations, the authors provided the ways of analyzing the areas where a loaded beam breaks away from the rigid foundation used.

Transverse vibration analysis was also conducted in works [4,5], where analytical solutions to vibrations of a beam supported on springs (rotational or translational) on both ends, placed on two-parameter elastic foundation, were presented. The paper provides an explicit conclusion that the frequency of a system mode vibrations increases with the rising stiffness of the foundation and support springs.

An issue of transverse vibrations of two-segment beams was presented, among others, in [6]. The systems analyzed were divided into three basic groups depending on the shape of the cross-sectional area. The values of three first frequencies of mode vibrations at various mounting conditions were indicated. An identical analysis of changes in mode vibration frequency was carried out in [7], where the systems of three and more segments were taken into consideration. In the case of a system built of any finite number of segments, with an added discrete element in the form of concentrated mass or translation spring, the properties of Green’s function [8] were used to determine the changes in mode values. Publications [9] dealt with systems of beams of a linearly variable cross-section, but only one of the main dimensions of the cross section was subject to change.

This paper presents theoretical and numerical tests on the stability and free vibrations of a curved beam with a variable cross section at vertical load. A curved beam of a variable cross-section, variable inertia moment, and variable stiffness along its length is examined. Motion equations and boundary conditions of analyzed systems are formulated on the grounds of total mechanical energy. The paper also includes the results of simulation tests on the course of changes in mode vibrations over the whole switch point structure: dimensionless load parameter–dimensionless mode frequency parameter. None of the publications mentioned take into consideration a beam with four simultaneously occurring variable quantities (variable cross-section, variable inertia moment, variable stiffness, and beam curve).

## 2. Structure Turnouts Railways

The turnout is one of the most important elements of the railway network: it is the element where railway disasters occur most often. There are two elements in the turnout which determine proper functioning of the railway turnout. These are a switch point and a frog [10,11] shown in Figure 1.

Turnouts for high-speed rail, *V* > 160 km/h, have curvature radii greater than 1000 m. In these turnouts, switch points have the following features; they are elastic elements with a permanently fixed radius of curvature—from the point of view of mechanics that means that it is a curved beam with a variable cross-section, variable moment of inertia, and variable stiffness. In turnouts with smaller curvature radii, these are usually simple elements connected to the rail with an articulated element. The examination of such a beam (for *R* ≥ 1200 m) requires mathematical modeling to take into account all the elements that occur in the real object. In these systems, there are a minimum of 3 adjusting closures, in which the holding force for each should be greater than 7.5 kN. A mathematical model of a straight beam with a variable cross-section can be found in [12]. The results of these considerations were used to describe the switch point with smaller radii [13,14,15].

The switch point allows rail vehicle movement to be diverted into a straight or closure rail. The mathematical modeling process of the switch point needs to include forces coming from subsequent wheel sets; for example, for a rail vehicle composed of two bogies during the passage of the vehicle through the switch point, there are four forces in the resulting sequence: between sets in the bogie and between sets in two further bogies. These elements are considered in the modeling of the switch point as a beam subjected to a variable load [16,17,18].

## 3. Parameters Characterizing the Switch Point

Considering the switch point as a trapezoidal curved beam (Figure 2), lying on a continuous elastic substrate, and subjected to a vertical load, which passes through its neutral axis, let *F* mark the shear force in the cross-section of the beam at a distance *x* from the origin of the coordinates, and *M* is the moment of external forces relative to this cross-section. By *F* we understand the sum of all vertical forces acting on the beam from the side of the cross-section on which the origin of the coordinates is located. In [19,20,21], one can find mathematical models describing the dynamics of a beam with a variable cross-section, but they lack the impulse of forces acting on the surface in the vertical direction (wheel pressure on the switch point) and horizontal (the force holding a switch drive) [22,23].

The initial assumptions were made to simplify the process of applying differential equations to the description of the movement of a curved beam with a variable cross-section, which is an element of a beam associated with two degrees of freedom. Thus, one rotary movement and one translational movement take place at both ends of the beam. The action of *F* force causes deformations (displacements) in the direction *x* of the narrower part of the switch point. In Figure 2, the beginning of the switch point is marked by *A*, the end by *B*. In turn, the section depth of the beam with a variable cross-section is indicated by *g_a_* and *g_b_*. The length of the beam is marked by *L* [24,25].

### Switch Point Geometry as a Beam with a Variable Cross-Section

As for the structure of the beam geometry, the following assumptions have been made.
The *y* and *z* axes are the main central axes of inertia of the cross-section.The cross-section and longitudinal section has variable quantities and its height h changes linearly along the beam (Figure 3).The initial load in the coordinate system has zero speed, which changes later along the beam.The beam stiffness of the constrained torsion is equal to zero.The bending stiffness with respect to the y axis is slightly greater than the stiffness with respect to the *z* axis.

In mathematical considerations, the geometrical parameters of the cross-section of the beam marked with the red envelope (Figure 4a), were used. The profiles change all the values describing the structure along the x-axis, which directly affects the change of the cross-sectional area of the beam. This, in turn, changes the moments of inertia occurring at individual sections defined in the *xyz* axes in Figure 4b.

According to the cross-section of the beam shown in Figure 4a, the dependence on the cross-section area [26,27] is determined as follows,
(1)A(x)=ac+be+d(x)g
where,
(2)$$d\left(x\right)=\frac{h_{2}-h_{1}}{L}x+h_{1}$$

The moment of inertia I in the x-axis is defined by
(3)$$I_{x}=\frac{1}{3}a^{3}c+\frac{1}{3}b^{3}e+\frac{1}{3}h_{1}g^{3}+\frac{h_{2}-h_{1}}{L}\cdot \frac{1}{3}g^{3}$$

The moment of inertia I in the z-axis is described by the following equation.
(4)$$\begin{array}{cc} I_{z}=\frac{1}{12}ac^{3}+ac & {\left(d\left(x\right)+e+\frac{c}{2}-\frac{\left[\frac{1}{2}be^{2}+a\left(d\left(x\right)+e+\frac{c}{2}\right)c+d\left(x\right)\left(\frac{d\left(x\right)}{2}+e\right)g\right]}{\left(be+ac+d\left(x\right)g\right)}\right)}^{2}\\ & +\frac{1}{12}be^{3}+be{\left(\frac{\left[\frac{1}{2}be^{2}+a\left(d\left(x\right)+e+\frac{c}{2}\right)c+d\left(x\right)\left(\frac{d\left(x\right)}{2}+e\right)g\right]}{\left(be+ac+d\left(x\right)g\right)}-\frac{e}{2}\right)}^{2}\\ & +\frac{1}{12}gd{\left(x\right)}^{3}\\ & +gd\left(x\right){\left(\frac{d\left(x\right)}{2}+e-\frac{\left[\frac{1}{2}be^{2}+a\left(d\left(x\right)+e+\frac{c}{2}\right)c+d\left(x\right)\left(\frac{d\left(x\right)}{2}+e\right)g\right]}{\left(be+ac+d\left(x\right)g\right)}\right)}^{2} \end{array}$$

The moment of inertia I in the y-axis was defined by the dependence
(5)$$I_{y}=\frac{1}{12}a^{3}c+\frac{1}{12}b^{3}e+\frac{1}{12}g^{3}d\left(x\right)$$

## 4. Beam with a Variable Cross-Section, Switch Point, Model of Assumption, and Real Object

For the switch point modeling process, the function characterizing the change in the quantities of moments of inertia along the length was determined. The results are shown in Figure 5.

These results were obtained based on measurements in the real object. The switch point shown in Figure 6a for a turnout with a radius of 1200 m will be shown as part of a curved beam. The assumption was that the switch point would be a curved beam of length L with a variable cross-section whose main axis runs along the central point of its cross-section [28,29]. Where the radius of curvature *R* defining the axis of the beam is large in relation to its section width *h*
$\left(\frac{R}{h}>5\right)$, it can be assumed that under working stresses of a cross-section defined in this way, the beams produce identical effects as in the case of a straight beam (R→∞). Therefore, in further considerations, it has been adopted that the curved beam meets the assumption $\left(\frac{R}{h}<5\right)$.

It was assumed that the radius would be larger than 1200 m and a mathematical description would be presented.

The switch point will be considered as a beam of a defined curvature and length *s*. The cross-section is a variable parameter in relation to its length in the planes *x_1_* and *x_3_* cut perpendicularly by two planes relative to the main axis. In the discussed model, *F(L)* indicated the forces acting on the beam (switch point), whereas all the moments of inertia appearing in the considered beam are marked by *m(L)*. Both the *F(L)* and the *m(L)* parameters are determined per unit length of the main axis of the beam [30,31,32].

The phenomena occurring on such a defined beam can be divided into three components: peripheral ($p_{\theta }$ circular force and circular moment of inertia $M_{\theta }$), radial (radial force $p_{\zeta }$ and moment of radial inertia $M_{\zeta }$), and normal (normal forces and moments of inertia). Figure 6b shows the model of a switch point defined by a curved beam with applied components. Forces and moments (normal, peripheral, and radial) acting on individual segments of the beam on the left side were marked by N, Q_2_, $Q_{\zeta }$ and $M_{\theta }$, $M_{2}$ i $M_{\zeta }$, respectively. For the right side, the individual components are defined as follows;N+ΔN, $Q_{2}+\Delta Q_{2}$, and $Q_{\zeta }+\Delta Q_{\zeta }$, and the moments are defined by values $M_{\theta }+\Delta M_{\theta }$, $M_{2}+\Delta M_{2}$, and $M_{\zeta }+\Delta M_{\zeta }$. The values in which the symbol Δ is displayed are considered variable. Parameter Δs determines the force vector in the switch point. As you can see in Figure 6b, these values will always be positive [33,34,35].

To obtain the state of balance in the analyzed model, the following equations must be satisfied,
(6)$$\sum F_{\theta }=-N\cos\left(\frac{\Delta \theta }{2}\right)+\left(N+\Delta N\right)\cos\left(\frac{\Delta \theta }{2}\right)-Q_{\zeta }\sin\left(\frac{\Delta \theta }{2}\right)-\left(Q_{\zeta }+\Delta Q_{\zeta }\right)\sin\left(\frac{\Delta \theta }{2}\right)+p_{\theta }\Delta s=0\;\sum F_{2}=-Q_{2}+\left(Q_{2}+\Delta Q_{2}\right)+p_{s}\Delta s=0\;\sum F_{\zeta }=N\sin\left(\frac{\Delta \theta }{2}\right)+\left(N+\Delta N\right)\sin\left(\frac{\Delta \theta }{2}\right)-Q_{\zeta }\cos\left(\frac{\Delta \theta }{2}\right)+\left(Q_{\zeta }+\Delta Q_{\zeta }\right)\cos\left(\frac{\Delta \theta }{2}\right)+p_{\zeta }\Delta s=0$$
and
(7)$$\begin{array}{cc} \sum M_{\theta }^{\left(0\right)}=-M_{\theta } & \cos\left(\frac{\Delta \theta }{2}\right)\\ & +\left(M_{\theta }+\Delta M_{\theta }\right)\cos\left(\frac{\Delta \theta }{2}\right)+m_{\theta }\Delta s\\ & -M_{\zeta }\sin\left(\frac{\Delta \theta }{2}\right)-\left(M_{\zeta }+\Delta M_{\zeta }\right)\sin\left(\frac{\Delta \theta }{2}\right)-Q_{2}R\cos\left(\frac{\Delta \theta }{2}\right)\\ & +\left(Q_{2}+\Delta Q_{2}\right)R\cos\left(\frac{\Delta \theta }{2}\right)+Rp_{2}\Delta s=0 \end{array}\;\sum M_{2}^{\left(0\right)}=-M_{2}+\left(M_{2}+\Delta M_{2}\right)+m_{2}\Delta s+R\left[N-\left(N+\Delta N\right)\right]-Rp_{\theta }\Delta s=0\;\begin{array}{cc} \sum M_{\zeta }^{\left(0\right)}=-M_{\zeta } & \cos\left(\frac{\Delta \theta }{2}\right)+\left(M_{\zeta }+\Delta M_{\zeta }\right)\cos\left(\frac{\Delta \theta }{2}\right)+m_{\zeta }\Delta s+M_{\theta }\sin\left(\frac{\Delta \theta }{2}\right)\\ & +\left(M_{\theta }+\Delta M_{\theta }\right)\sin\left(\frac{\Delta \theta }{2}\right)+Q_{2}R\sin\left(\frac{\Delta \theta }{2}\right)+\left(Q_{2}+\Delta Q_{2}\right)R\sin\left(\frac{\Delta \theta }{2}\right)R=0 \end{array}$$

Assuming that
(8)$$\underset{s\to 0}{\lim}\frac{\Delta \theta }{\Delta s}=\frac{1}{R}$$
and
(9)$$\underset{\theta \to 0}{\lim}\left[\frac{\sin\left(\frac{\Delta \theta }{2}\right)}{\left(\frac{\Delta \theta }{2}\right)}\right]=1$$
where *R(L)* is the radius of the switch point from the axis of beam curvature. Dividing expressions (6) and (7) by Δs at the limit Δs→0 and θ→0 allowed obtaining dependencies so that the moments of inertia of the switch point $M_{\zeta }$, $M_{z}$ were determined. Therefore, the following equations are true [36],
(10)$$\frac{dN}{ds}-\frac{Q_{\zeta }}{ds}+p_{\theta }=0\;\frac{dQ_{2}}{ds}+p_{2}=0\;\frac{N}{R}+\frac{dQ_{\zeta }}{ds}+p_{\theta }=0$$
and
(11)$$\frac{dM_{\theta }}{ds}-\frac{M_{\zeta }}{R}+m_{\theta }+R\left(\frac{dQ_{2}}{ds}+p_{2}\right)=0\;\frac{dM_{2}}{ds}-R\left(\frac{dN}{ds}+p_{\theta }\right)=0\;\frac{dM_{\zeta }}{ds}+\frac{M_{\theta }}{R}+Q_{2}=0$$

Considering the mutual relations between Equations (10) and (11), we obtained [37,38]
(12)$$\frac{dM_{\theta }}{ds}-\frac{M_{\zeta }}{R}+m_{\theta }=0\;\frac{dM_{2}}{ds}-Q_{\zeta }=0\;\frac{dM_{\zeta }}{ds}+\frac{M_{\theta }}{R}+Q_{2}=0$$

Expressions (11) and (12) determine the state of balance for a flat switch point. For a straight beam →∞, $s\to x_{1}$, in turn the indexes ζ→3 and thus θ→1. From the equations presented above, the moment of inertia $M_{2}$ described by the equation was determined [39,40,41]:(13)$$\frac{d^{3}M_{2}}{ds^{3}}+\frac{R^{\prime }}{R}\cdot \frac{d^{2}M_{2}}{ds^{2}}+\frac{1}{R^{2}}\cdot \frac{dM_{2}}{ds}+\frac{R^{\prime }}{R}p_{\zeta }+\frac{dp_{\zeta }}{ds}-\frac{p_{\theta }}{R}=0$$
where
(14)$$R^{\prime }=\frac{dR}{ds}$$

What is more, from the same relationships, you can get expressions for moments $M_{\theta }$ and $M_{\zeta }$ described as follows [42,43,44],
(15)$$\frac{dM_{\theta }}{ds}-\frac{M_{\zeta }}{R}+m_{\theta }=0\;\frac{d^{2}M_{\zeta }}{ds^{2}}-\frac{R^{\prime }M_{\theta }}{R^{2}}+\frac{1}{R}\frac{dM_{\theta }}{ds}-p_{2}=0$$

According to the analysis carried out in the case of a switch point, individual moments of inertia are given directly from curved places and in the case of a straight rail at its ends. It should be added that the switch point is exposed to the same phenomena as the straight rail. Therefore, we decided to carry out an analysis determining the bending effect of the beam. Differential equations that describe the dynamics of the switch point movement (beam with a variable cross-section) are solved to obtain the required stiffness of the surface and beam deformations, and can also be used to estimate the mode vibrations described by the mathematical equations of the object [45,46,47].

To determine motion equations for a curved beam with a variable cross-section, first, the model of the tested object should be correctly defined. For this purpose, a curved beam with a variable cross-section and with the vertical force from the wheelset operating on its upper surface is considered. This situation is illustrated in Figure 7, where the value of u means deflections of the beam in three directions: longitudinal, vertical, and transverse. The radius determining the beam curvature is defined by *R*, and its length by *L*. Among *x* and *y* axes, which coincide with the main axis of the system defining the cross-section of the tested element, the *x*-axis is tangent to the switch point arc. Dislocations (deflections) *u_x_*, *u_y_*, and *u_z_* of the center of gravity should be determined for each section of the beam cross-section. The sizes *θ_x_*, *θ_y_*, and *θ_z_* describe rotations around the three axes [48,49,50].

In further considerations, the accepted principle is that all deformations occurring in the analyzed model are small, so in further mathematical analysis a linear theory was used. It was also assumed that the beam has very little deformation resistance. Thus, differential equations defining the movement of a curved beam with a variable cross-section are as follows.

The movement specifying an axial shift:(16)$$E\cdot A_{A}{\left[1+\left(\frac{d_{B}}{d_{A}}\right)\cdot \frac{x}{l}\right]}^{m}\cdot \left(\frac{\partial^{2}u_{x}}{\partial x^{2}}+\frac{1}{R}\frac{\partial u_{z}}{\partial x}\right)=0$$

The movement specifying radial dislocation:(17)$$EI_{y}\left(\frac{\partial^{4}u_{z}}{\partial x^{4}}+\frac{2}{R^{2}}\cdot \frac{\partial^{2}u_{z}}{\partial x^{2}}+\frac{u_{z}}{R^{4}}\right)+\frac{E\cdot A_{A}}{R}{\left[1+\left(\frac{d_{B}}{d_{A}}\right)\cdot \frac{x}{l}\right]}^{m}\cdot \left(\frac{\partial^{2}u_{x}}{\partial x^{2}}+\frac{1}{R}\frac{\partial u_{z}}{\partial x}\right)=0$$

The movement defining a vertical displacement of a curved beam with a variable cross-section:(18)$$EI_{z}\left(\frac{\partial^{4}u_{y}}{\partial x^{4}}+\frac{1}{R}\cdot \frac{\partial^{2}\theta_{x}}{\partial x^{2}}\right)-\frac{GJ}{R}\left(\frac{\partial^{2}\theta_{x}}{\partial x^{2}}+\frac{1}{R}\frac{\partial^{2}u_{y}}{\partial x^{2}}\right)=0$$

The motion defining the torsional movement:(19)$$\frac{E\cdot I_{z}}{R}\left(\frac{\partial^{2}u_{y}}{\partial x^{2}}-\frac{\theta_{x}}{R}\right)+GJ\left(\frac{\partial^{2}\theta_{x}}{\partial x^{2}}+\frac{1}{R}\frac{\partial^{2}u_{y}}{\partial x^{2}}\right)=0$$
where the entire description of the movement dynamics of the curved beam with a variable cross-section is shown for the longitudinal axis x, the quantities *E* and *G* denote, respectively, the moduli of elasticity of the beam material and its stiffness. *A*: cross-sectional area, and *I_y_* and *I_z_* are, respectively, moments of inertia determined around the *y*-axis and *z*. *J*: constant torsional. From the deductions, note that the differential equations determining the movement for the plane of displacements *u_x_* and *u_z_* are independent of the displacements occurring outside the plane *u_y_* and $\theta_{x}$. In addition, the differential equations determined for the axial displacement *u_x_* and for the radial shift *u_z_* are reciprocally incorporated, and the situation looks the same for the equation determining the vertical displacements of *u_y_* and torsional movement. Next, it is assumed that both ends of the beam are supported, which translates into the lack of dynamics of the movement of the flexible displacement and the rotational movement. Thus, their first derivatives are not “zero”. In general, this can be triggered by the action of a rail vehicle traveling on a beam. In mathematical considerations, the vehicle movement may be replaced by a vertical moving force as shown in Figure 6.

The movement in the horizontal direction along a curved beam with a variable cross-section on the upper surface of the tested element is considered as a perpendicular force F moving at *v* velocity and acting on the upper surface of the beam. Load of the object moving on the beam is expressed by *F_0_ = −mg*, where g is gravitational acceleration. The equations of vertical vibrations of a curved beam with a variable cross-section are determined based on the dependence (16–19). However, in this case, to better reflect the actual conditions in modeling the phenomena of beam motion dynamics, nonlinearity is taken into account.
(20)$$m\frac{\partial^{2}u_{y}}{\partial x^{2}}+EI_{z}\left(\frac{\partial^{4}u_{y}}{\partial x^{4}}-\frac{1}{R}\frac{\partial^{2}\theta_{x}}{\partial x^{2}}\right)-\frac{GJ}{R}\left(\frac{\partial^{2}\theta_{x}}{\partial x^{2}}+\frac{1}{R}\frac{\partial^{2}u_{y}}{\partial x^{2}}\right)=F_{0}\delta \left(x-vt\right)\;\rho J\frac{\partial^{2}\theta_{x}}{\partial x^{2}}+\frac{EI_{z}}{R}\left(\frac{d^{2}u_{y}}{dx^{2}}-\frac{\theta_{x}}{R}\right)+GJ\left(\frac{d^{4}\theta_{x}}{dx^{4}}+\frac{1}{R}\frac{d^{2}u_{y}}{dx^{2}}\right)=0$$
where *m* stands for the mass per unit length, ρ is the density of the curved beam, and δ(t) is the Dirac impulse.

It can be seen that the right side of the first Equation (16) can be expressed as the sum of a series of sinusoidal functions that satisfy the boundary conditions:(21)$$mu_{y}\left(x,t\right)=\overset{\infty }{\underset{i=1}{\sum }}q_{yi}\left(t\right)\sin\frac{j\pi x}{L}$$
where $q_{yi}$ means the generalized coordinate of the tested beam section for the i-th note. The expression of the steering angle is marked by $\theta_{x}$ and must be defined by the Equation (16). Subsequently, by replacing Equation (17) with the last Equation (18) and taking into account the boundary conditions regarding the angle of twist of the beam, we obtained
(22)$$\theta_{x}\left(x,t\right)=\overset{\infty }{\underset{i=1}{\sum }}\gamma_{i}q_{yi}\left(t\right)\sin\frac{j\pi x}{L}$$
where the parameter $\gamma_{i}=-R{\left(\frac{j\pi }{L}\right)}^{2}\cdot \frac{\left(GJ+EI_{z}\right)}{\left[{\left(\frac{j\pi }{L}\right)}^{2}GJR^{2}+EI_{z}\right]}$; then, as a result of the mutual relationship, we obtained
(23)$$\theta_{x}\left(x,t\right)=\overset{\infty }{\underset{i=1}{\sum }}q_{0i}\left(t\right)\sin\frac{j\pi x}{L}$$
where $q_{0i}$ means *i*-th coordinate for the steering angle $\theta_{x}$.

Further considerations refer to defining the deflection of the beam and determining the turning angle:(24)$$u_{y}\left(x,t\right)=q_{y1}\sin\frac{\pi x}{L},\;\;\;\theta_{x}\left(x,t\right)=q_{01}\sin\frac{\pi x}{L}$$
where by $q_{y1}$ and $q_{01}$, respectively, the first generalized coordinates for displacement $u_{y}$ and steering angle $\theta_{x}$ have been determined.

The nominal model used to determine the vibrations of a curved beam with a variable cross-section is shown in Figure 8.

To solve the differential equations presented in (16), the Galerkin method should be used. Thus, we multiply both sides of the first equation by $\delta u_{y}$ variable and the second equation by $\delta \theta_{x}$ variable. Then, we apply approximation to the first form for quantity $u_{y}$ and $\theta_{x}$ in Equation (3.18). In the next step, we perform the operation of combining the two differential equations, each with respect to *x* in the range from *0* to *L*. The result of this operation is the mathematical relationships:(25)$$\left(\frac{\partial^{2}q_{y1}}{\partial x^{2}}+a_{1}q_{y1}+a_{2}q_{y1}\right)\delta q_{y1}=\frac{2F_{0}}{mL}\sin\frac{\pi vt}{L}\delta q_{y1}\;\left(\frac{\partial^{2}q_{\theta 1}}{\partial x^{2}}+b_{1}q_{01}+b_{2}q_{y1}\right)\delta q_{01}=0$$
where
(26)$$a_{1}=-\frac{1}{\rho J}\left[\frac{EI_{z}}{R}+GJ{\left(\frac{\pi }{L}\right)}^{2}\right]\;a_{2}=-\frac{1}{\rho J}\frac{1}{R}{\left(\frac{\pi }{L}\right)}^{2}\left[EI_{z}+GJ\right]\;b_{1}=\frac{1}{m}{\left(\frac{\pi }{L}\right)}^{2}\left[\frac{EI_{z}}{R}{\left(\frac{\pi }{L}\right)}^{2}+\frac{GJ}{R^{2}}\right]\;b_{2}=\frac{1}{mR}{\left(\frac{\pi }{L}\right)}^{2}\left[EI_{z}+GJ\right]$$

Because variables $\delta q_{y1}$ and $\delta q_{01}$ are arbitrary, Equation (25) is reduced to the following,
(27)$$\frac{\partial^{2}q_{y1}}{\partial x^{2}}+a_{1}q_{y1}+a_{2}q_{y1}=\frac{2F_{0}}{mL}\sin\frac{\pi vt}{L}\;\frac{\partial^{2}q_{\theta 1}}{\partial x^{2}}+b_{1}q_{01}+b_{2}q_{y1}=0$$

The general solution to the Equation (27) which is a homogeneous one is
(28)$$q_{y1}=q_{y1h}+q_{y1p}\;q_{01}=q_{01h}+q_{01p}$$
where indices h and p denote the solution of Equation (27), respectively.

Thus, the solutions can be presented as follows,
(29)$$q_{y1h}=h_{1}\sin\omega_{v1}t+h_{2}\csc\omega_{v1}t\;q_{01h}=k_{1}\sin\omega_{v1}t+k_{2}\csc\omega_{v1}t$$
where $\omega_{v1}$ is the basic vibration frequency determined in the vertical direction of a curved beam with a variable cross-section, and the parameters $h_{1},h_{2},k_{1},k_{2}$ are, respectively, coefficients that should be determined on the basis of the initial boundary conditions. In the next step, on the basis of the expression (25), which is substituted into Equation (27), the F_0_ value was determined:(30)$$\left[\begin{array}{cc} a_{1}-\omega_{v1}^{2} & a_{2}\\ b_{2} & b_{1}-\omega_{v1}^{2} \end{array}\right]\cdot \left[\begin{array}{c} q_{y1h}\\ q_{01h} \end{array}\right]=\left\{\begin{array}{c} 0\\ 0 \end{array}\right\}$$

Then, by performing an operation of substituting the determinant equal to zero into Equation (24), the fundamental frequency $\omega_{v1}$ can be determined from the relationship
(31)$$\omega_{v1}=\sqrt{\frac{a_{1}+b_{1}+\sqrt{{\left(a_{1}-b_{1}\right)}^{2}+4a_{2}b_{2}}}{2}}$$

To obtain individual higher harmonic frequencies, it can be assumed that
(32)$$\begin{array}{c} q_{y1h}=p_{y1}\sin\frac{\pi vt}{L}\\ q_{01h}=p_{01}\sin\frac{\pi vt}{L} \end{array}$$
where $p_{y1}$ and $p_{01}$ are the value of the amplitude of vibrations occurring in a given object. Next, by carrying out the substitution (32) to Equation (27) the following was obtained,
(33)$$\left[\begin{array}{cc} a_{1}-{\left(\frac{\pi v}{L}\right)}^{2} & a_{2}\\ b_{2} & b_{1}-{\left(\frac{\pi v}{L}\right)}^{2} \end{array}\right]\cdot \left[\begin{array}{c} p_{y1}\\ p_{01} \end{array}\right]=\left\{\begin{array}{c} \frac{2F_{0}}{mL}\\ 0 \end{array}\right\}$$

From Equation (33), the solution for $p_{y1}$ was determined:(34)$$p_{y1}=\frac{2P_{0}}{mL}\cdot \frac{1}{\omega_{v1}^{2}}\frac{1}{1-S_{v1}^{2}}$$
where
(35)$$S_{v1}=\frac{\pi v}{L\omega_{v1}},\;\;\;\beta =\frac{b_{1}-{\left(\frac{\pi v}{L}\right)}^{2}}{b_{1}+a_{1}-\omega_{v1}^{2}-{\left(\frac{\pi v}{L}\right)}^{2}}$$
where $S_{v1}$ specifies the velocity parameter for vertical vibrations occurring on a curved beam, which is defined as the ratio of the frequency of generated vibrations $\frac{\pi v}{L}$ transmitted by the load to the fundamental frequency $\omega_{v1}$.
(36)$$q_{y1}\left(t\right)=p_{y1}\left(\sin\frac{\pi vt}{L}-S_{v1}\sin\omega_{v1}t\right)$$

In a further stage of mathematical considerations it was assumed that the initial displacement and the speed of the load are zero if the object does not move. Such conditions should be taken into account when solving the differential equations defined in (28) or performing summation of solutions of equations (27) and (29). The coefficients in the solution defined by $h_{1}=-p_{1}S_{v1}$ and $h_{2}=0$ result from the solution of the coefficient $q_{y1}$ from
(37)$$q_{y1}\left(t\right)=p_{y1}\left(\sin\frac{\pi vt}{L}-S_{v1}\sin\omega_{v1}t\right)$$

According to the assumption that $F_{0}=-m_{v}g$ and Equation (34), the above equation can be rewritten as
(38)$$q_{y1}\left(t\right)=-\frac{2m_{v}g}{mL}\cdot \frac{1}{\omega_{v1}^{2}}\cdot \frac{1}{1-S_{v1}^{2}}\cdot \beta \psi_{v1}\left(t\right)$$
where the amplitude function $\psi_{v1}\left(t\right)$ is determined by
(39)$$\psi_{v1}\left(t\right)=\sin\frac{\pi vt}{L}-S_{v1}\sin\omega_{v1}t$$

As a result, vertical displacements of the curved beam with a variable cross-section are expressed by the equation
(40)$$u_{y}\left(x,t\right)=-\frac{2m_{v}g}{mL}\cdot \frac{1}{\omega_{v1}^{2}}\cdot \frac{1}{1-S_{v1}^{2}}\cdot \beta \psi_{v1}\left(t\right)\cdot \sin\frac{\pi vt}{L}$$

For the middle section $x=\;\frac{L}{2}$ of a curved beam with a variable cross-section, the vertical displacements are:(41)$$u_{y}\left(\frac{L}{2},t\right)=\frac{2m_{v}g}{mL}\cdot \frac{1}{\omega_{v1}^{2}}\cdot \frac{1}{1-S_{v1}^{2}}\cdot \beta \psi_{v1}\left(t\right)$$

The presented mathematical analysis, which aimed to determine the equations of the movements for a curved beam with a variable cross-section refers only to obtaining a solution that takes into account the adopted boundary conditions. In practice, the load caused by a rail vehicle passing through the switch point is so short that it induces a temporary process. The results regarding higher vibration frequencies occurring on the tested element depend on the location of the load on the switch point and the speed of the rail vehicle. In the further part of the article, results will be presented in a form of higher harmonic vibrations for a 1200-m turnout.

## 5. Results of the Simulation Study

The numerical solutions were obtained with the finite element calculation program-ANSYS, which allows modeling the behavior of structural systems, while taking into account the diverse but complementary material properties of beam objects in the area of construction and mutual cooperation of these construction materials. Computer simulations of this type contribute to the reduction of costs associated with conducting experimental design tests. They do not replace these tests but allow more effective planning of experiments. They enable the analysis of complex issues of beam behavior during loading in the entire strain range from the linear–elastic range, through the elastic deformation phases in the areas of tensile stress causing initial and advanced scratch states, and in areas of compressive stress causing switch point deflections that lead to the destruction of the railway turnout construction elements.

Based on the presented nominal models, mathematical models were determined and calculations were made for a beam having a turnout switch point with a radius of 1200 m, and the forces that arise when the rail vehicle travels through the turnout on stock and closure rail were taken into account. The results of simulation tests were performed in the ANSYS software.

For computer simulations, a curved beam with a variable cross-section in the shape of a rail with the profile: 49 E1 (S49), 60E1 (UIC60) of steel grade 900A was used. Technical data of the UIC60 profile can be found in publication [51]. In the numerical model under study, rounding and chamfering of the switch point were omitted to reduce the calculation time. On the curved beam model with a variable cross-section, a tetra-type finite element grid of 24082 elements was applied. The vibration analysis was obtained by combining strength and frequency analysis. The results of calculations from the strength analysis were implemented as input data for frequency analysis [52]. To induce mechanical loads, the model was fixed on the left side and the supports were added, one fixed and one movable. The parameters used in the simulation are shown in Table 1.

The calculations were carried out in one bending case:Bending in the direction of (x axis). The beam is curved in the plane of action of the concentrated force with the value F_0_ = 12 kN. The force is applied at the middle connector of the top edge of the free end of the beam and works in the direction of the x axis (negative). Theoretical value of the maximum displacement in this direction according to the work is-0.2623 cm.

Simulation research was also carried out based on work [52]. Own forms were determined: first, second, third, and fourth for x equal to the length of the switch point, for x = 0 the switch point is free (Figure 9). The simulation results were determined for different nominal forces affecting the switch points. Figure 10 and Figure 11 show the designated own forms for a curved beam with variable stiffness (change of section area, stiffness and moments of inertia) characterizing the properties of the switch point.

The purpose of the calculations was to determine the vibrations caused by the action of a 12 kN load. Figure 10 shows the waveforms of individual harmonics (from the first to the sixth one) obtained from computer simulations. However, the distribution of beam deflections is shown in Figure 11. The obtained results of vibrations and deflections indicate the influence of the load on the formation of strains and displacements in the analyzed case. The numerical simulation performed with the ANSYS program allowed us to determine displacements at the load acting on the upper surface of the curved beam with a variable cross-section. The maximum deflection of the beam, i.e., elastic displacement in the x-axis direction, occurs at the end of the tested element and amounts to 1.1398 mm, whereas the maximum longitudinal displacements of the beam with a value of 3.2194 mm appear at the end of it.

## 6. Conclusions

The numerical analysis carried out in this work made it possible to determine the displacements and vibrations of higher harmonics in the model of a curved beam with variable cross-section subjected to load. The presented method can be used for testing the dynamics of a rail vehicle moving on the switch point with variable cross-section, variable stiffness, and variable inertia moment for greater radii of curvature. The results obtained in the conducted simulations enable the comparison of the switch point mode frequency with own vibration frequency of a rail vehicle mass, which may lead to the creation of dynamic couplings between the turnout and the rail vehicle (Medelsztam theory).

The presented results of the numerical analysis were obtained in the ANSYS 16 Workbench module.

Solving continuous beams with analytical methods is to a large extent complicated and time-consuming, whereas ANSYS enables quick numerical calculations using FEM (Finite Element Method).

The conducted numerical tests allowed concluding that greater system stiffness and the stiffness of applied Winkler foundation result in a higher frequency of system mode vibration. According to Figure 9 diagrams, the frequency of transverse vibrations depends mainly on the applied support. Also, a generated vertical force and a very location of a segment of a curved beam with variable parameters in relation to supports significantly affect vibration frequency.

In certain cases, the equations derived may be used for the verification of computer calculations and, in the authors’ opinion, have some educational value. They may also be applied in calculations of other structures whose shape corresponds to a linear or quadratic relation between the inertia moment and the length of a beam.

The approximation of variable parameter beam shape affects the mode vibration frequency value depending on parameters that describe a mechanical system—a switch point and values of a load acting on it. A change of cross-section and beam curve may result either in an increase or a decrease of mode vibration frequency values. The maximum load value being equal to zero value of basic mode vibration frequency depends mainly on the value of parameters describing a beam’s shape, geometrical parameters and its load structure. Maximum forces obtained from the kinetic criterion of stability loss are reflected in the results of numerical analyses of the system, which were conducted on the grounds of the static criterion of stability loss.

In further research work, the authors will focus on testing the impact of vertical force on the possibility of modification of transverse vibration frequencies when other support techniques are used and a perpendicular force from a switch drive is considered.

## Figures and Tables

**Figure 1 materials-13-00701-f001:**
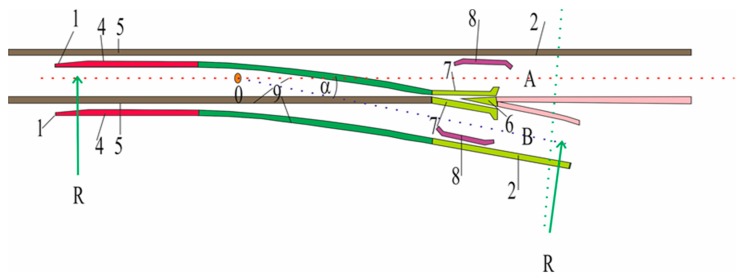
The basic components and geometrical elements of the regular turnout are: beginning of turnout 1 (in pre-switch contact point); end of turnout 2 (in contact point behind the frog); turnout geometric center, 0, which is the intersection of the stock rail axis with the closure rail axis; radius *R* of the closure rail curve; turnout angle α (angle between the axes of the stock rail and the closure rail), slant of the turnout—tangent α—expressed as a vulgar fraction with one in the numerator; switch points, 4; rheostats, 5; frog, 6; wing rails, 7; guard rails, 8; connecting rails, 9; AOB triangle.

**Figure 2 materials-13-00701-f002:**
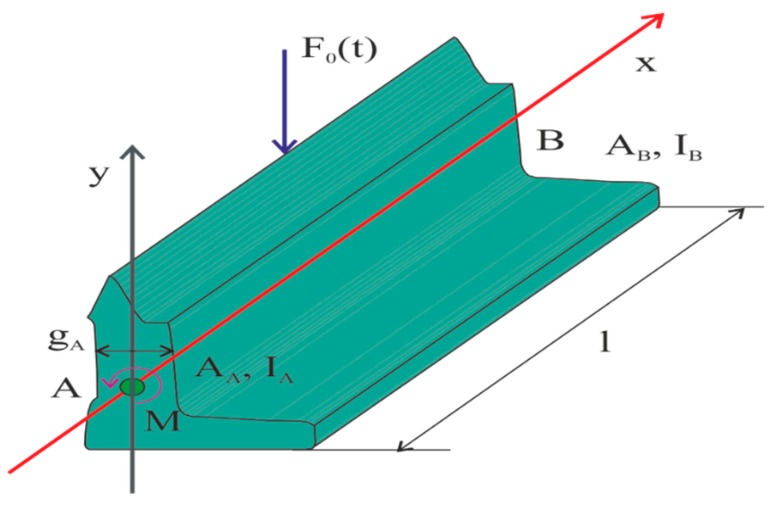
Parameters of the switch point.

**Figure 3 materials-13-00701-f003:**
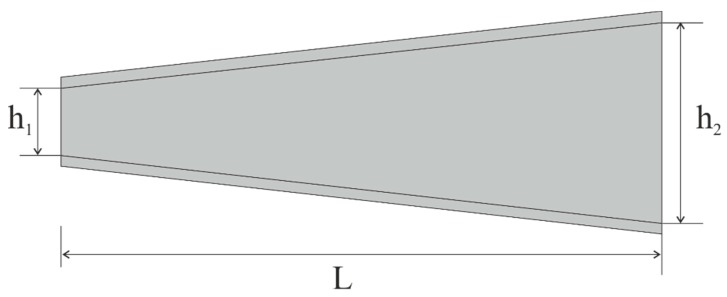
Top view of the beam used for numerical analysis.

**Figure 4 materials-13-00701-f004:**
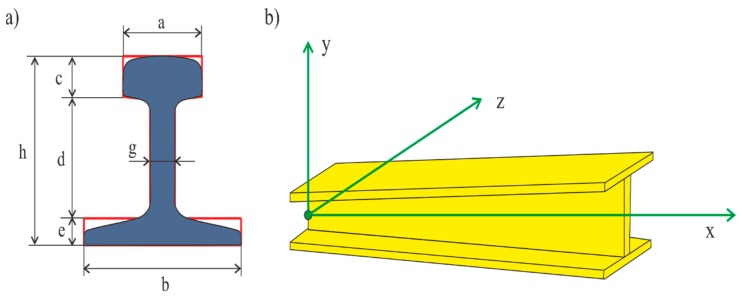
Parameters of the beam cross-section and position of the beam in the global coordinate system, (**a**) forward view, (**b**) side view.

**Figure 5 materials-13-00701-f005:**
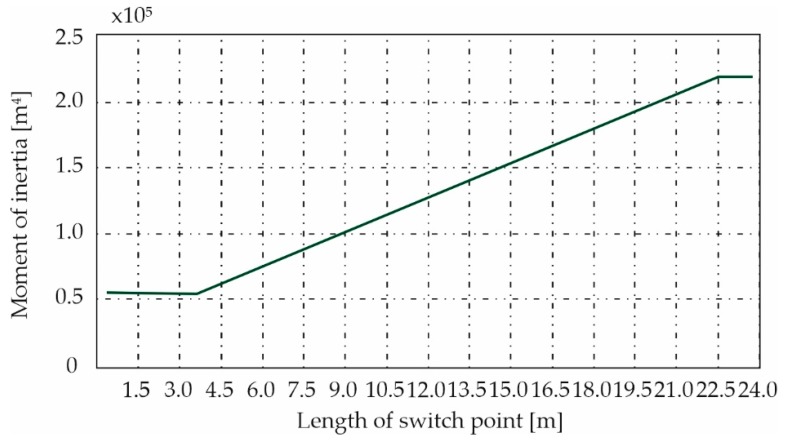
Diagram of the moment of inertia change for a beam with a variable cross-section.

**Figure 6 materials-13-00701-f006:**
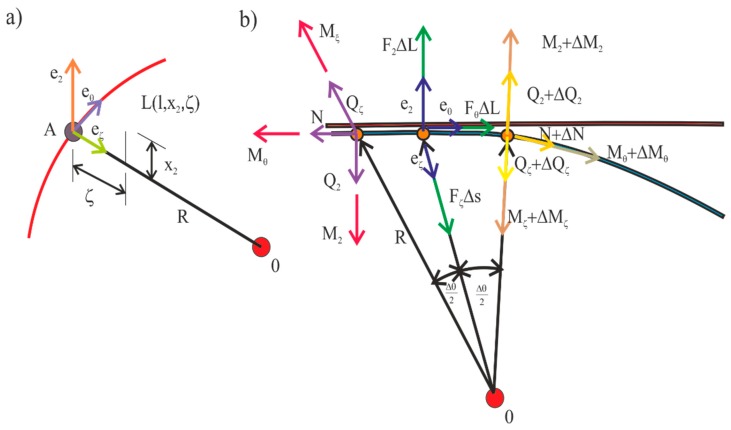
Coordinate system defining the switch point (**a**) and model of the switch point (**b**).

**Figure 7 materials-13-00701-f007:**
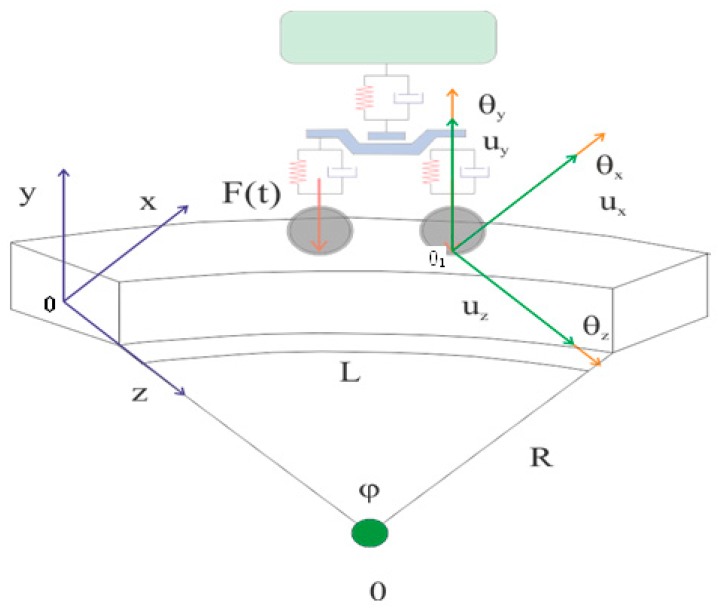
Model of a curved beam with a variable cross-section used in mathematical modeling, *0_1_* - contact point between the wheel and the rail.

**Figure 8 materials-13-00701-f008:**
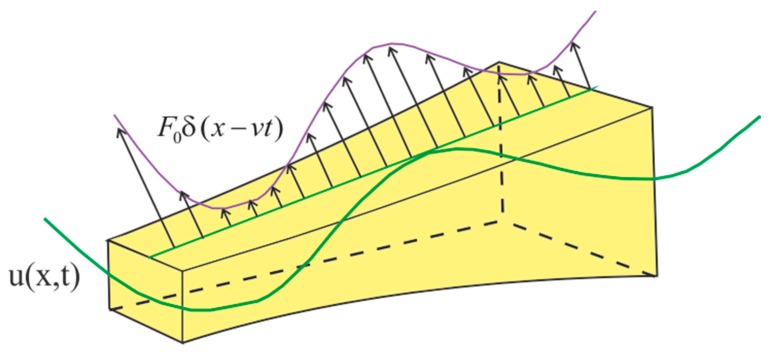
The nominal model of the beam used to determine vibrations.

**Figure 9 materials-13-00701-f009:**
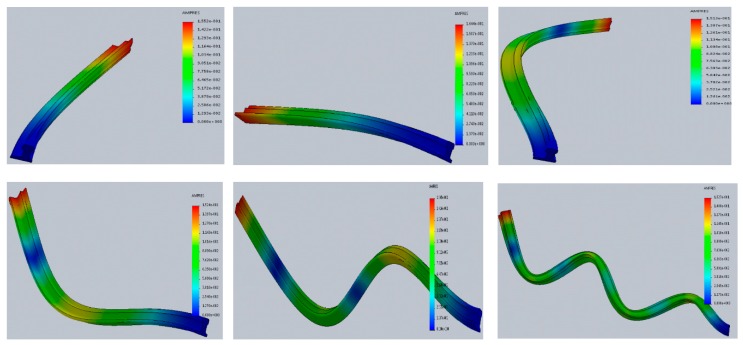
Mode vibrations of the beam with a variable cross-section (from the first natural frequency to the sixth—from the left side).

**Figure 10 materials-13-00701-f010:**
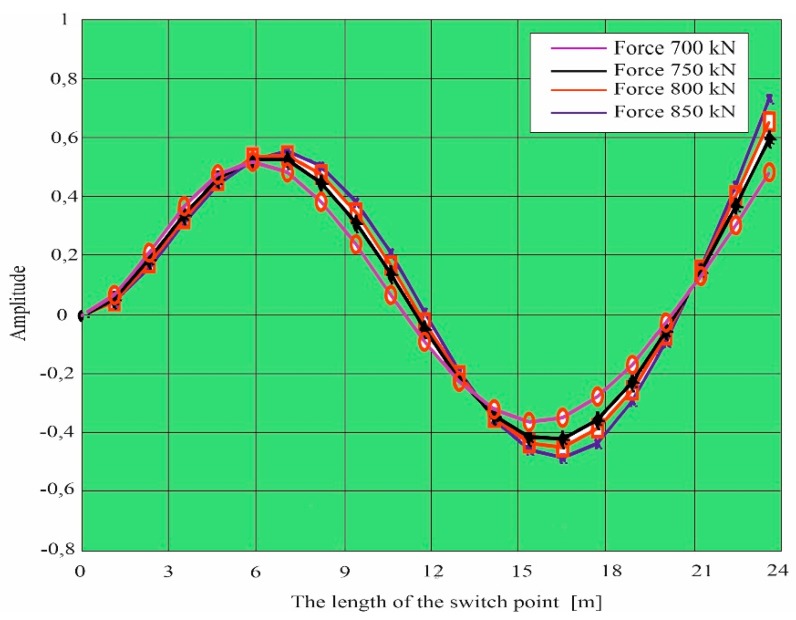
Third own form for a curved beam with a variable cross-section (turnout switch point).

**Figure 11 materials-13-00701-f011:**
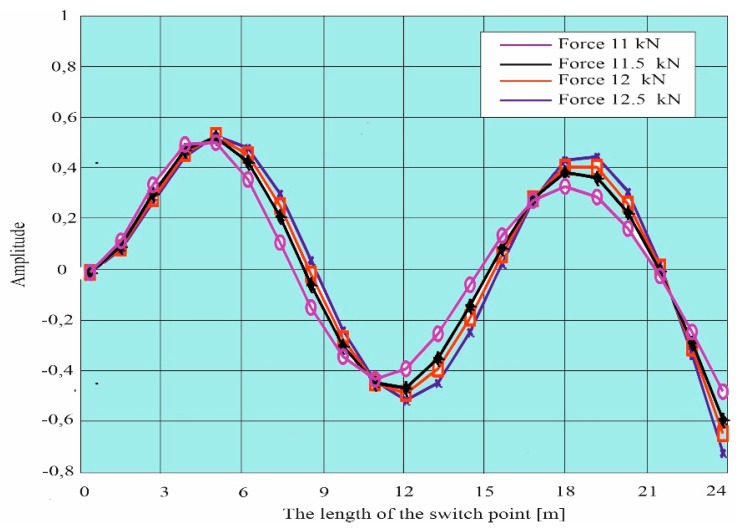
The fourth own form for a curved beam with a variable cross-section (turnout switch point).

**Table 1 materials-13-00701-t001:** Switch point parameters used in simulations.

Name	Symbol	Value
Vertical force F_0_	F_0_	12 kN
Coefficient of substrate elasticity	k	3600 Nm^2^
Young’s modulus	E	$2.1\times 10^{11}\mathrm{Pa}$
Poisson’s ratio	v	0.3
Density	ρ	7800 kg/m^3^

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
