# Peer review of "Numerical Testing of Switch Point Dynamics—A Curved Beam with a Variable Cross-Section"

_materials, 2020, doi:10.3390/ma13030701_

Round 1

Reviewer 1 Report

-Abstract: The text must be carefully revised. Some sentences contain mistakes (in the abstract: very general statements) whereas some sentences must be reworded as the English is “meaningless”.

Line number: 17 - '....normal force reflecting forces...'.

Line number 18-19: 23 m  and 1200 m. No need to represent the dimension units in parenthesis.

Line number 20: need to be rephrased.

I strongly recommend that the authors retain the services of a professional editor. There are many reputable companies that offer these services.

Introduction is poorly written. Proper references need to be used rather than using others. This need to be taken care. Language can be improved. The sentences are half constructed or incomplete in a way that the readers are expected to fend for themselves in order to understand their meaning. 

Section 3 heading is misleading. Figure 5 is poorly represented. Need to be corrected. Axis format is bad.

The theory is well known and it found in many classical books. There are too many equations which are repeated from literature. If it is not derived by authors, it is better to curtail few equations.

Overall the paper appears to be of satisfactory quality. Only if the language was better, it would require lesser effort to understand and comprehend what the writer intends to convey. I would recommend this paper to be published after minor revisions.

Author Response

Thank you for your review and all your comments, which will be taken into account in further scientific work.
All formal comments have been included in the revised text.
We have the following opinion on the theoretical comment. We believe that there has been a misunderstanding, because the theory concerning straight beams and fixed cross-sections (constant stiffness and constant moment of inertia is known and this is the right comment of the Reviewer). In the publication, the authors deal with a beam with a variable cross-section along the OX axis, which changes the stiffness and moment of inertia, and the beam is curved. There is no such case with these four variable sizes in the literature, so it required full derivation of the equations. The study of such a case is not known to the authors of the publication. The cited literature refers only to methods of solving partial differential equations.

Reviewer 2 Report

COMMENTS TO AUTHORS

The work developed in the document is not a research process as such. The Authors have verified the usefulness of ANSYS simulation software for the study of physical and mechanical phenomena associated with the movement of trains on railroad tracks.

The work developed in the document is not a research process as such. The Authors have verified the usefulness of XXX simulation software for the study of physical and mechanical phenomena associated with the movement of trains on railroad tracks.

However, the document shows an experience of application of the ANSYS software that may be of interest to professionals in the calculation of structures. I consider that the work developed is related to the editorial objectives of Materials.

The Authors have used an appropriate methodology for the research objectives. However, it is necessary to make some comments about it:

Abstract

The text must be reviewed and completed. The Authors should include at the end of the text, by way of conclusion, a brief comment on the validity of the experimental process followed.

References

The bibliography used in the research is appropriate, but the authors must make an exhaustive review of the way in which they have shown the bibliographic references. References to journals should be made with their abbreviated title and DOI (Digital Object Identifier) must be added.

Authors should follow the indications of the model used by the MDPI Editorial, which can be found on the Materials website

Conclusions

The conclusions of the research carried out are very elementary and are limited to indicating the validity of the ANSYS program in the study of the efforts that occur in dynamic couplings between the turnout and the rail vehicle.

Taking as reference the knowledge about the finite element mathematical model, the Authors should make a more extensive and precise comparative study with the results obtained from the simulation with the ANSYS program.

Final comment

I congratulate the Authors for the work done and encourage them to continue their researches.

Author Response

Thank you for your review and all the substantive comments that will be taken into account in further scientific work.
The authors do not share the opinion that the article is not a research task, because its aim is to show what kind of phenomena can occur in a beam, where there are four types of variable values: change of section, change of stiffness, change of moment of inertia and curvature. This required the formulation of a mathematical model and the study of the behaviour of this type of beam. The results obtained are the basis for further research related to the movement of a rail vehicle on this type of beam (turnout switch point of 1200 m radius). The tool used was selected because of its availability. The presented testing process can be used for beams - deitch point with larger radius, e.g. R = 3000 m or R = 10000 m, which are used in turnouts for high-speed railway.
The summary has been corrected according to the notes.
The bibliography has been completed and a DOI added.
The conclusions were supplemented by the possibility of dynamic couplings between the vehicle and the turnout - switch point - beam (according to Mandelstam theory). Such an approach was not found in any publication.
From the point of view of mechanics, a turnout is a slightly different system than a beam. It has a number of "imperfections", e.g. a switch point or a crossing point.